# Does 3D-Assisted Operative Treatment of Pelvic Ring Injuries Improve Patient Outcome?—A Systematic Review of the Literature

**DOI:** 10.3390/jpm11090930

**Published:** 2021-09-18

**Authors:** Hester Banierink, Anne M. L. Meesters, Kaj ten Duis, Job N. Doornberg, Mostafa El Moumni, Erik Heineman, Inge H. F. Reininga, Frank F. A. IJpma

**Affiliations:** 1Department of Trauma Surgery, University Medical Center Groningen, University of Groningen, 9713 GZ Groningen, The Netherlands; a.m.l.meesters@umcg.nl (A.M.L.M.); k.ten.duis@umcg.nl (K.t.D.); j.n.doornberg@umcg.nl (J.N.D.); m.el.moumni@umcg.nl (M.E.M.); i.h.f.reininga@umcg.nl (I.H.F.R.); f.f.a.ijpma@umcg.nl (F.F.A.I.); 2Department of Orthopedics, University Medical Center Groningen, University of Groningen, 9713 GZ Groningen, The Netherlands; 3Department of Surgery, University Medical Center Groningen, University of Groningen, 9713 GZ Groningen, The Netherlands; e.heineman@umcg.nl

**Keywords:** pelvic ring injury, sacroiliac screw, three-dimensional, 3D virtual surgical planning, 3D printing, navigation

## Abstract

**Background:** There has been an exponential growth in the use of advanced technologies for three-dimensional (3D) virtual pre- and intra-operative planning of pelvic ring injury surgery but potential benefits remain unclear. The purpose of this study was to evaluate differences in intra- and post-operative results between 3D and conventional (2D) surgery. **Methods:** A systematic review was performed including published studies between 1 January 2010 and 22 May 2020 on all available 3D techniques in pelvic ring injury surgery. Studies were assessed for their methodological quality according to the Modified McMaster Critical Review form. Differences in operation time, blood loss, fluoroscopy time, screw malposition rate, fracture reduction and functional outcome between 3D-assisted and conventional (2D) pelvic injury treatment were evaluated and a best-evidence synthesis was performed. **Results:** Eighteen studies fulfilled the inclusion criteria, evaluating a total of 988 patients. Overall quality was moderate. Regarding intra-operative results of 3D-assisted versus conventional surgery: The weighted mean operation time per screw was 43 min versus 52 min; for overall operation time 126 min versus 141 min; blood loss 275 ± 197 mL versus 549 ± 404 mL; fluoroscopy time 74 s versus 125 s and fluoroscopy frequency 29 ± 4 versus 63 ± 3. In terms of post-operative outcomes of 3D-assisted versus conventional surgery: weighted mean screw malposition rate was 8% versus 18%; quality of fracture reduction measured by the total excellent/good rate by Matta was 86% versus 82% and Majeed excellent/good rate 88% versus 83%. **Conclusion:** The 3D-assisted surgery technologies seem to have a positive effect on operation time, blood loss, fluoroscopy dose, time and frequency as well as accuracy of screw placement. No improvement in clinical outcome in terms of fracture reduction and functional outcome has been established so far. Due to a wide range of methodological quality and heterogeneity between the included studies, results should be interpreted with caution.

## 1. Introduction

Pelvic ring injuries have an estimated annual incidence of 14–37 per 100,000 inhabitants each year [1,2]. Treatment can be either non-operative or operative, depending on the injury as well as patient characteristics. The operative treatment of pelvic ring injuries remains a challenging task for surgeons due to the complex three-dimensional (3D) shape of the pelvis, morphological variations, limited access to fracture sites, and narrow bone corridors for screw placement [3]. The goal of operative treatment is to restore pelvic symmetry and achieve stable fracture fixation, which allows for early mobilization and good functional outcome at the long-term [4,5]. Progress in 3D imaging technologies has resulted in an exponential increase in the usage of these techniques—that is both industry- as well as surgeon-driven- for preoperative planning and for translation of the plan to the operative procedure [6]. In essence, 3D-assisted surgery encompasses a wide spectrum of modalities including 3D virtual preoperative planning, 3D-printed models for pre-contouring of osteosynthesis plates and 3D navigational tools. Some coin these 3D (printing) techniques the “second industrial revolution” in *Orthopaedic Trauma Surgery*. Nevertheless, the additional clinical value of 3D techniques in pelvic surgery has yet to be elucidated, both practically as well as scientifically.

Conventional X-rays and two-dimensional (2D) computed tomography (CT) images are to date widely used to assess fracture characteristics, reduction quality and positions of osteosynthesis materials in pelvic ring injury treatment [3]. However, 3D virtual models may allow the surgeon to gain more insight in the fracture pattern, surgical approach, and positions of osteosynthesis materials. It has been reported that pre-operative virtual simulation and 3D printing-assisted pre-contoured plate fixation of pelvic ring injuries resulted in precise pre-operative planning and accurate execution of the operative procedures [3]. Moreover, 3D-assisted surgery for percutaneous screw placement may lower the risk of complications and decrease the need for revision surgery due to a lower rate of screw malposition [7]. However, there is a lack of studies with sufficient statistical power to provide evidence on superiority of the available 3D technologies compared to conventional (2D) techniques in different types of pelvic ring injuries.

Hence, the main objective of the present systematic review was to analyse differences in outcomes between currently available 3D-assisted and conventional (2D) pelvic ring injury treatment. Therefore, we asked (1) What is the difference in intra-operative results in terms of operation time, blood loss, screw malposition and fluoroscopy time between 3D-assisted and conventional (2D) surgery? and (2) What is the difference in post-operative results in terms of fracture reduction and functional outcome between 3D assisted and conventional (2D) surgery?

## 2. Methods

This systematic review was performed according to the Preferred Reporting Items for Systematic Reviews (PRISMA) [8]. The review protocol has been registered in PROSPERO International prospective register of systematic reviews under registration number CRD42021224915.

### 2.1. Identification of Studies: Search Strategy

The MEDLINE-Pubmed and Ovid-EMBASE libraries were searched on May 22nd of 2020 for articles published between 1 January 2010 until 22 May 2020. The search string was developed in collaboration with an experienced medical librarian (Table 1). It was developed to identify references related to 3D-imaging and 3D-operative techniques of pelvic ring injuries. Therefore, the items “pelvis”, “injury” and “3D/threedimension” were combined to develop the search strategy.

### 2.2. Inclusion and Exclusion Criteria

Eligible studies for inclusion reported either on (1) the use of 3D techniques in the virtual planning of operative treatment of pelvic ring injuries; (2) 3D printed templates with fracture visualization; (3) 3D printed templates with pre-operative plate contouring; (4) 3D virtual planning of screw trajectories; (5) 3D custom-made implants with guides; and (6) 3D navigation for screw placement. Patients should be 18 years or older and patients with fragility fractures of the pelvis (FFP) were included as well. Outcomes directly related to the operative treatment should be reported. These included operation time, blood loss, screw malposition rate and fluoroscopy time or fluoroscopy frequency, fracture reduction and functional outcome. These outcome measures represent the efficiency and accuracy of the surgical procedure itself. We hypothesized that 3D assisted surgery could have an effect on these measures, which is the rationale to choose these outcome measures. Moreover, these are widely used for assessing pelvic ring surgery related to patient outcomes [9,10,11]. Except for case studies with N < 10 and conference abstracts, all study designs were accepted for inclusion. Concerning language, studies written in English, German, Spanish, French and Dutch were included. Biomechanical and animal studies were excluded, as well as studies about classification of injuries by means of 3D techniques. Moreover, studies that included outcomes after both pelvic ring injuries and acetabular fractures and that did not differentiate between these injuries in terms of outcomes were excluded.

### 2.3. Study Selection

All articles were imported into Rayyan QCRI, a web-based sorting tool for systematic literature reviews [12]. The study selection was performed in two screening phases: (1) title and abstract screening, and (2) full text screening. Both selection phases were independently performed by the same researchers. Disagreement was resolved by discussion. The initial searches (conducted from 1 January 2010 to 22 May 2020) generated 819 articles and after removal of duplicates, 709 potential eligible studies were screened. Following title and abstract assessment, 34 articles were reviewed in full text. A total of 18 articles were included in the review of which most were case-control studies (N = 9), followed by cross-sectional cohort studies (N = 8) and one prospective cohort study. No randomized controlled trials (RCTs) were found in this search. Figure 1 demonstrates a flowchart of the inclusion procedure.

### 2.4. Data Extraction

The data extraction was independently conducted (HB, FIJ) using a precompiled extraction file (Microsoft Excel version 14.0; Microsoft Inc., Redmond, WA, USA). Study characteristics, fracture classification, 3D technologies and outcome measures were extracted from all the included studies by the senior author.

### 2.5. Assessment of Methodological Quality

Methodological quality and risk of bias of the included studies was independently assessed according to the guidelines of the McMaster University Occupational Therapy Evidence-Based Practice Research Group [13]. The Modified McMaster Critical Review form consists of nine categories: citation, study purpose, literature, design, sample, outcomes, intervention, results, and conclusions and implications. This review form is appropriate to assess RCTs, cohort studies, single-case designs, before- and after-designs, case control studies, cross-sectional studies and case studies. The guidelines established by Law et al. [13] were utilized for the quality assessment. Every item was answered with ‘yes; 1 point’, ‘no; 0 points’, ‘not addressed; 0 points’ or ‘not applicable (N/A); no points given’. Any continued disagreements were solved during a consensus meeting (HB and FIJ). The total score reflects the methodological quality with a maximum score of 17 for RCTs and 13 for other designs. The definitive score is calculated in a percentage and may vary from 0–100%, with a higher score indicating a higher methodological quality. Scores below <60% were considered as poor quality, scores between 60–74% were considered as moderate quality, scores between 75–89% indicated good-quality and scores between 90–100% indicated excellent-quality studies.

The results of the quality assessment of the included articles are presented in Table 2. A maximum score of 13 could be obtained as four items concerning RCTs were left out. Total scores in percentages ranged between 46% and 92% with a mean score of 63% (SD 16). Only one study was considered as excellent quality, five were good-quality, four moderate quality and eight poor quality studies.

### 2.6. Outcomes

Outcomes relevant to the operation were recorded. These parameters included operation time, blood loss, screw malposition (varying from contacting cortical bone to actual perforation of the cortical bone), fluoroscopy dose, amount and frequency, fracture reduction according to the guidelines established by Tornetta and Matta [29] and patient- or physician-reported functional outcome.

### 2.7. Patient and Injury Characteristics

Overall, data of a total of 988 patients were reported in the studies (Table 3). Most studies (N = 12) focused on unstable pelvic ring injuries (Type B and Type C according to the AO classification system) [30]. Of all included patients, 694 received 3D-assisted pelvic ring injury surgery and 294 had conventional surgery. The study characteristics are shown in Table 3.

### 2.8. Strategy for Data Synthesis and Statistical Analysis

Data synthesis involved the comparison, combination, and summary of findings. Data are presented as part of a narrative synthesis, involving text and tables. Continuous variables are presented as means with standard deviation (SD) (parametric data) or as median with interquartile range (IQR) in case of non-parametric data. Dichotomous variables are given as frequency and percentages. Due to the retrospective nature of the included studies and the heterogeneity of their design, results could not be pooled for statistical analysis. Instead, weighted means of the various outcome variables of the studies were calculated for comparison. Besides, a best-evidence synthesis was performed, taking into account the methodological quality and outcome of the original studies (Table 4) [31]. Excellent and good quality studies were labeled as high-quality studies whereas moderate and low-quality studies were labeled as low-quality.

## 3. Results

### 3.1. Intra-Operative Results

Our first question asks about the difference in intra-operative results in terms of operation time, blood loss and fluoroscopy time between 3D-assisted and conventional (2D) surgery. All identified 3D-assisted surgery techniques are shown in Figure 2 and described in Appendix A.

#### 3.1.1. Operation Time per Screw and Overall Operation Time

Two out of three case-control studies reported that 3D-assisted surgery led to a significant decrease in operation time per screw using 3D printed drilling guides [16] and intra-operative 3D imaging [17] (Table 5). Berger-Groch et al. [14] found no difference using intra-operative 3D imaging. Weighted mean operation time per screw was 15 min (range 14–18) for 3D-assisted and 26 min (range 19–40) for the conventional group.

Three case-control studies reported on a significant decrease in overall operation time using a 3D printed model [4,19] and a 3D printed model combined with pre-contouring of the osteosynthesis plate [3] compared to the conventional technique. The other case-control study found no difference using intra-operative 3D imaging for screw placement [28]. Weighted mean overall operation time was 97 min (range 59–206) for 3D-assisted and 113 min (range 72–276) for the conventional group.

#### 3.1.2. Blood Loss

The only case-control study by Hung et al. [3] reported a significant decrease in blood loss (275 ± 197 mL versus 549 ± 404 mL; *p* = 0.023) using pre-operative virtual simulation and 3D printing-assisted plate contouring. No weighted mean blood loss of all studies was calculated as both open and percutaneous surgery was applied.

#### 3.1.3. Fluoroscopy Dose, Time and Frequency

One case-control study by Yang et al. [16] reported a significantly decreased fluoroscopy dose by using 3D printed drilling guides in comparison to conventional surgery. The other case-control study by Beck et al. [22] did not express the difference in fluoroscopy dose in a *p*-value. No weighted mean could be calculated because different units were used to report dose.

Three of the four case-control studies reported that intra-operative 3D-assisted surgery significantly reduced fluoroscopy time [14,17,28]. The remaining case-control study by Beck et al. [22] did not express the difference in fluoroscopy time with a *p*-value. Weighted mean fluoroscopy time was 74 s (range 22–29) for the 3D-assisted and 125 s (range 58–248) in the conventional group.

One case-control study by Cai et al. [4], combining 3D visualization with a 3D printed model for screw placement, reported a significant decrease in fluoroscopy frequency (29 ± 4 versus 63 ± 3; *p* < 0.001) compared to the conventional technique.

### 3.2. Post-Operative Results

Our second question asks about the difference in post-operative results in terms of fracture reduction and functional outcome.

#### 3.2.1. Screw Malposition

Two out of five case-control studies reported significantly less screw malposition rates by using a 3D printed drilling [16] and intra-operative 3D image guided surgery [17]. Two other studies found no difference using intra-operative 3D image guided surgery [14,18]. Beck et al. [22] did not report on the difference expressed by a *p*-value. Weighted mean screw malposition rate for 3D-assisted surgeries was 8% (range 0–22.6) compared to 18% (range 5–24) in the conventional group (varying from contacting cortical bone to actual perforation of the cortical bone).

#### 3.2.2. Post-Operative Reduction Score

Two case-control studies did not report an improved quality of the reduction of the fracture by using a 3D printed model [4] or a 3D printed drilling guide [16]. Weighted mean reduction score was 86% (range 79–100) for 3D-assisted surgery and 82% (range 81–89) in the conventional group according to the Tornetta and Matta criteria [29].

#### 3.2.3. Functional Outcome

The two-case control-studies did not report an increase in functional outcome using a 3D printed model [4] or intra-operative 3D imaging [17]. Weighted mean rate of the Majeed score “excellent” and “good” for 3D-assisted surgery was 88% (range 82–100) and 83% (range 81–89) in the conventional group.

### 3.3. Best-Evidence Synthesis

#### 3.3.1. Intra-Operative Results

Compared to conventional pelvic ring injury surgery, moderate evidence was found for a decrease in operation time per screw, operation time overall, blood loss, fluoroscopy dose and fluoroscopy time. The evidence for a decrease in fluoroscopy frequency was limited. Conflicting evidence was found for a decrease in screw malposition rate.

#### 3.3.2. Post-Operative Results

Moderate evidence was found that fracture reduction as well as functional outcome did not improve using 3D assisted pelvic ring injury surgery.

## 4. Discussion

No overview exists on the currently available 3D technologies and to what extent they contribute to the operative treatment of pelvic ring injuries. In this systematic review we evaluated outcomes of the complete spectrum of innovative 3D technologies applied for pelvic ring injury surgery over the past decade. Thereby, it provides a clinically question-driven overview about the ongoing debate whether these advanced 3D technologies contribute to the results of operations and patient recovery. It encompasses 18 articles, showing that previously applied 3D-assisted pelvic ring injury surgery can be divided in five main groups. These include ‘3D virtual fracture visualization and preoperative planning’, ‘3D printed model assisted surgery’, ‘pre-contouring of osteosynthesis material’, ‘3D printed surgical guides’, and ‘intra-operative 3D imaging’. The results reveal that the application of these technologies seem to have a positive effect on the operative treatment of pelvic ring injuries by shortening the duration of surgery, decreasing blood loss as well as fluoroscopy frequency, dose and time and minimizing risks on screw malposition. No difference in fracture reduction and functional outcome between 3D-assisted and conventional surgery was established.

Limitations of this systematic review are considered small patient groups and a wide range of methodological quality including a substantial number of moderate and poor-quality studies. Hence, a best-evidence synthesis was performed which is a transparent and commonly applied method attempting to answer the key questions [31,32]. Moreover, high heterogeneity between the studies was observed in terms of different outcome variables used. As a result, a limited number of comparative studies addressed all outcome variables of interest.

Our first question concerned the effects of 3D-assisted surgery on intra-operative outcomes including operation time, blood loss, fluoroscopy time, dose and frequency as well as screw malposition. The results of this systematic review reveal some potential intra-operative advantages by using 3D-assisted surgery. Overall, operative time can be reduced by using 3D printed models. This is in line with a meta-analysis performed by Zhang et al. [33]. Additionally, operation time per screw is shown to be decreased using 3D navigation in percutaneous sacroiliac screw placement. One case-control study showed that blood loss might be reduced by using 3D printing assisted contoured template compared to conventional surgery [3]. Fluoroscopy time can be effectively reduced by using 3D techniques as shown by three case-control studies [14,17,28]. Moreover, fluoroscopy dose and frequency might be reduced, although more studies are needed to actually draw conclusions with regard to these outcome measures. The majority of the articles (13 out of 18) in our systematic review reported on use of 3D navigation for percutaneous screw placement. Based on these results, we may cautiously conclude that 3D navigation tends towards a decrease in screw malposition, although larger comparative studies are needed. This is in line with the systematic review and meta-analysis by Zwingmann et al. [10].

Our second research question concerned the effects of 3D-assisted surgery on post-operative outcomes including fracture reduction and functional outcome. According to the reduction score by Tornetta and Matta, no difference could be found by using 3D-assisted surgery in comparison with conventional surgery. However, to date reduction measurements in pelvic radiographs have not been validated and interobserver reliability has shown to be poor [34]. Moreover, no evidence for improved functional outcome was found using 3D-assisted surgery for pelvic ring injuries. Nonetheless, only a limited number of studies with different methodological quality reported on these outcome measures after iliosacral screw fixation. Hence, future high-quality comparative studies on all five 3D techniques are needed to clarify whether post-operative reduction and functional outcome may benefit from 3D-assisted surgery.

## 5. Conclusions

Overall, five different techniques of 3D-assisted surgery were identified and are currently in use for pelvic ring injury treatment. These included ‘3D virtual fracture visualization and preoperative planning’, ‘3D printed model assisted surgery’, ‘pre-contouring of osteosynthesis material’, ‘3D printed surgical guides’, and ‘intra-operative 3D imaging’. These 3D-based techniques offer additional tools to improve intra-operative efficiency in terms of operation time, blood loss, fluoroscopy dose, time and frequency as well as accuracy of screw placement. However, improved anatomical reduction or functional outcome following 3D-assisted surgery has not been established so far. Due to the heterogeneity of the included studies in terms of methodological quality and number of studies that evaluated each of the outcomes of interest, results should be interpreted with caution. Future high-quality comparative studies are necessary to further establish possible advantages of 3D-assisted surgery in the treatment of pelvic ring injuries.

## Figures and Tables

**Figure 1 jpm-11-00930-f001:**
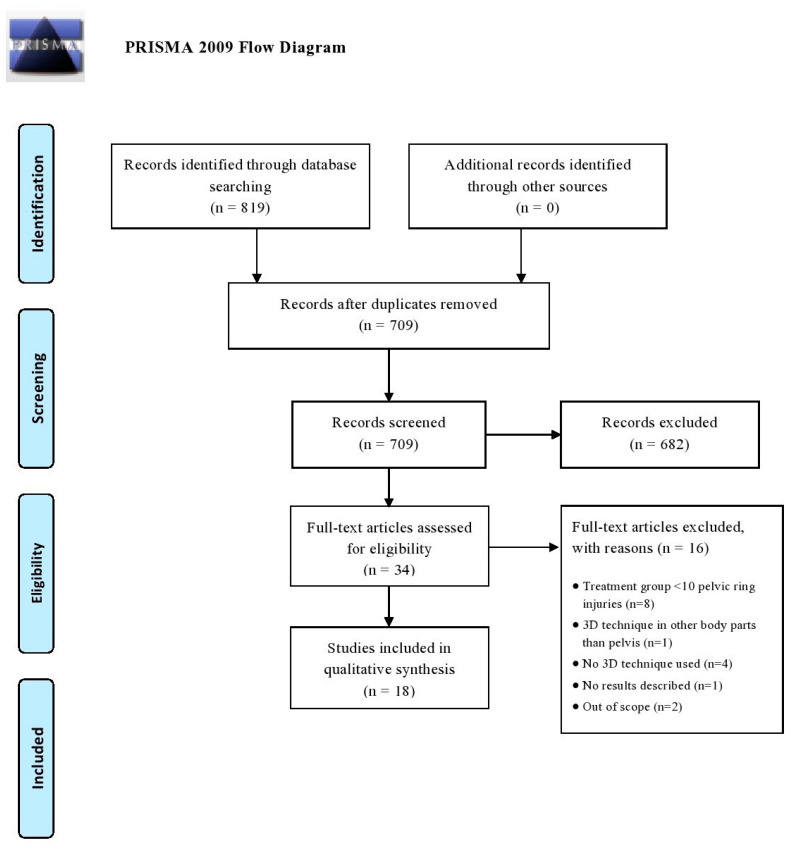
Flow diagram according to the PRISMA method.

**Figure 2 jpm-11-00930-f002:**
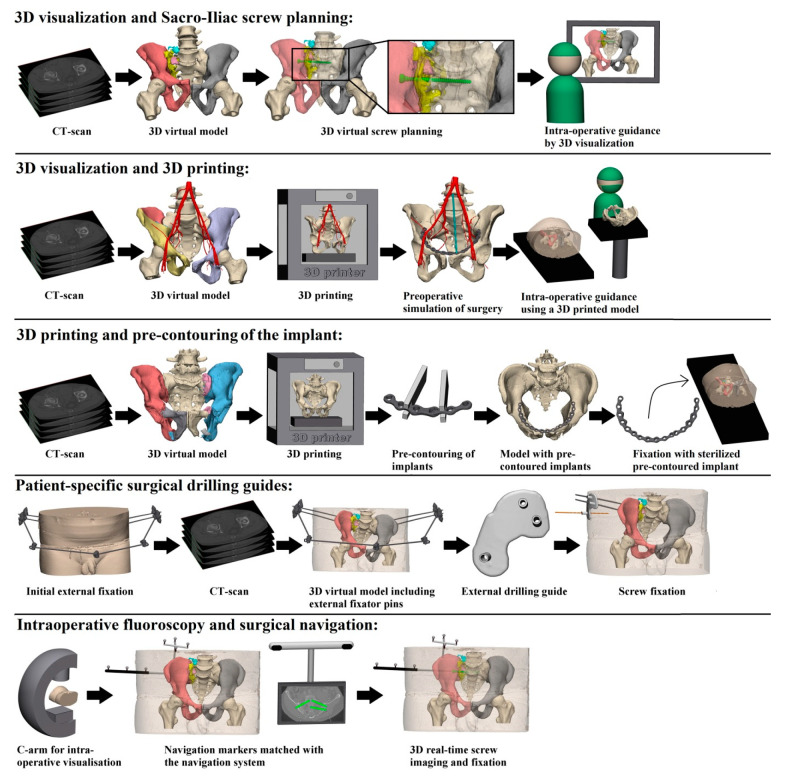
Presentation of the five identified 3D-assisted surgery techniques.

**Table 1 jpm-11-00930-t001:** Search strings by database.

Database	Search String
MEDLINE-PubMed	(((“Pelvis”[Mesh] OR pelvic ring[tiab]) AND (“Wounds and Injuries”[Mesh] OR “injuries” [Subheading] OR injur*[tiab] OR fractur*[tiab]))) AND ((3D[tiab] OR three dimension*[tiab] OR 3 dimension*[tiab] OR “Printing, Three-Dimensional”[Mesh] OR “Imaging, Three-Dimensional”[Mesh] OR navigation[tiab])) AND 2010:2020[dp]
Ovid-EMBASE	(‘pelvis’/exp OR ‘pelvis surgery’/exp OR ‘pelvic ring’:ti,ab) AND (‘bone injury’/exp OR injur*:ti,ab OR fractur*:ti,ab) AND (‘three dimensional printing’/exp OR ‘three-dimensional imaging’/exp OR 3d:ti,ab OR ‘three dimension*’:ti,ab OR ‘3 dimension*’:ti,ab OR navigation:ti,ab) AND [embase]/lim AND (2010–2020)/py

**Table 2 jpm-11-00930-t002:** Scores of the quality assessment list ranged from best to worst score.

−	1	2	3	4	5	6	7	8	9	10	11	12	13	Total	%
	Citation	Study Purpose	Literature Review	Sample		Outcomes		Intervention	Results				Conclusion and Clinical Implications		
Berger-Groch et al. [14]	+	+	+	+	−	+	+	+	+	+	+	+	+	12/13	92
Takao et al. [15]	+	+	+	+	−	+	+	+	+	+	+	−	+	11/13	85
Yang et al. [16]	+	+	+	+	−	+	+	+	+	+	+	−	+	11/13	85
Hung et al. [3]	+	+	+	+	−	+	−	+	+	+	+	−	+	10/13	76
Li. et al. [17]	+	+	+	+	−	+	−	+	+	+	+	−	+	10/13	76
Teo et al. [18]	+	+	+	+	−	+	+	+	+	−	+	−	+	10/13	76
Cai et al. [4]	+	+	+	+	−	−	−	+	+	+	+	−	+	9/13	69
Li et al. [19]	+	+	+	+	−	−	−	+	+	+	+	−	+	9/13	69
Balling [6]	+	+	+	+	−	+	+	+	−	−	+	−	−	8/13	62
Takeba et al. [20]	+	+	+	+	−	+	+	+	−	−	+	−	−	8/13	62
Pieske et al. [21]	+	+	+	+	−	−	−	+	+	−	+	−	−	7/13	54
Beck et al. [22]	+	+	+	+	−	−	−	+	−	−	+	−	−	6/13	46
Chen et al. [23]	+	+	+	+	−	−	−	+	−	−	+	−	−	6/13	46
Gao et al. [24]	+	+	+	+	−	−	−	+	−	−	+	−	−	6/13	46
Ghisla et al. [25]	+	+	+	+	−	−	−	+	−	−	+	−	−	6/13	46
Kim et al. [26]	+	+	+	+	−	−	−	+	−	−	+	−	−	6/13	46
Nie et al. [27]	+	−	+	+	−	−	−	+	+	−	+	−	−	6/13	46
Privalov et al. [28]	+	+	+	−	−	−	−	+	+	−	+	−	−	6/13	46

Every plus (+) sign means that the question was answered with ‘yes’. Every minus (−) sign means that a question was answered with ‘no’ or ‘not addressed’. The final two columns represent the total scores and percentages of maximal attainable scores (%).

**Table 3 jpm-11-00930-t003:** Study characteristics.

No.	Study	Year	N	Method *	Study Period	Injury Type	Intervention
1	Balling [6]	2019	52	CSS	2011–2016	Sacral FFPs^®^	3D image guided sacral screw fixation via single-sided minimally invasive transgluteal approach
2	Beck et al. [22]	2010	26	CCS	2008–2009	AO/Tile B, C	S: intra-operative 3D fluoroscopy of iliosacral screws and lumbopelvic implants (N = 14) C: iliosacral screws and lumbopelvic implants without intra-operative 3D (N = 12)
3	Berger-Groch et al. [14]	2018	136	CCS	2004–2014	AO/Tile B, C	S: 3D navigated iliosacral screw placement (N = 100) C: conventional iliosacral screw placement (N = 36)
4	Cai et al. [4]	2018	137	CCS	2014–2016	AO/Tile B, C	S: 3D printing-based minimally invasive cannulated screw treatment (N = 65) C: conventional surgery without 3D printing (N = 72)
5	Chen et al. [23]	2019	28	PCS	2016–2018	AO/Tile B, C	Minimally invasive screw fixation using the “Blunt End” Kirschner wire technique assisted by 3D printed external template
6	Gao et al. [24]	2011	22	CSS	2006–2008	AO/Tile B, C	Minimally invasive fluoro-navigation screw fixation
7	Ghisla et al. [25]	2018	21	CSS	2008–2017	Posterior pelvic ring	Intra-operative 3D-CT guided navigation for iliosacral screws
8	Hung et al. [3]	2018	30	CCS	2012–2017	AO/Tile A, B, C	S: ORIF with pre-operative virtual simulation and 3D- printing-assisted contoured plate (N = 16) C: ORIF with conventional plate fixation (N = 14)
9	Kim et al. [26]	2013	29	CSS	2010	AO/Tile A, B	Percutaneous iliosacral screwing using 3D-fluoroscopy
10	Li et al. [19]	2015	157	CCS	2009–2014	AO/Tile C	S: computer-aided angiography and rapid prototyping technology (N = 81) C: conventional imaging (N = 76)
11	Li. et al. [17]	2015	81	CCS	2005–2011	AO/Tile B, C	S: 3D C-arm fluoroscopy navigation (N = 43) C: C-arm fluoroscopy (N = 38)
12	Nie et al. [27]	2018	30	CSS	2015–2017	AO/Tile B, C	3D printing assisted by minimally invasive surgery for pubic rami fractures
13	Pieske et al. [21]	2015	71	CSS	Unknown	AO/Tile B, C	CT-guided sacroiliac percutaneous screw placement
14	Privalov et al. [28]	2020	53	CCS	2017–2018	Posterior pelvic ring	S: intra-operative CT in navigated sacroiliac instrumentation (N = 25) C^1^: navigated surgery with intra-operative 3D-C-Arm (N = 15) C^2^: conventional surgery with intra-operative control by 3D-C-Arm (N = 9) C^3^: conventional surgery with intra-operative control by 2D fluoroscopy (N = 4)
15	Takao et al. [15]	2019	27	CSS	2011–2016	AO/Tile B, C	3D fluoroscopic navigation of iliosacral screw insertion
16	Takeba et al. [20]	2018	10	CSS	2013–2017	AO/Tile B, C	O-arm and stealth station navigation for screw fixation
17	Teo et al. [18]	2018	36	CCS	2011–2016	AO/Tile B, C	S: sacroiliac screw placement with intra-operative navigation C: sacroiliac screw placement without intra-operative navigation
18	Yang et al. [16]	2018	40	CCS	2016–2017	AO/Tile B, C	S: 3D printed external template to guide iliosacral screw insertion (N = 22) C: conventional without external template (N = 18)

***** CSS, cross-sectional study; PCS, prospective cohort study; CCS, case-control study; S, study group; C, control group ^®^ FFP, fragility fracture of the pelvis.

**Table 4 jpm-11-00930-t004:** Best-evidence synthesis.

Best-Evidence Synthesis	
Strong evidence	Consistent findings among multiple high-quality studies
Moderate evidence	Consistent findings in multiple low-quality studies and/or one high-quality study
Limited evidence	Consistent findings in at least one low-quality study
Conflicting evidence	Inconsistent findings among multiple studies (high- and/or low-quality studies)
No evidence	Findings of eligible studies do not meet the criteria for one of the levels of evidence stated above, or there are no eligible studies available

**Table 5 jpm-11-00930-t005:** Study outcomes.

Measure	Study	3D Technology	Groups (N)		Outcomes		
			3D	Conventional	3D	Conventional	*p*-Value
**Intra-Operative Results**							
Operation time per screw (min) Mean ± std or Mean ± (range)	Berger-Groch et al. [14]	3D navigated iliosacral screw placement	100	36	48 ± 25	50 ± 29	0.74
	Chen et al. [23]	Minimally invasive screw fixation using the “Blunt End” kirschner wire technique assisted by 3D printed external template	28	-	21 ± 3	-	-
	Gao et al. [24]	Minimally invasive fluoro-navigation screw fixation	22	-	24 (16–45)	-	-
	Kim et al. [26]	Percutaneous iliosacral screwing using 3D-fluoroscopy	29	-	36 (18–83)	-	-
	Li. et al. [17]	Percutaneous screw fixation using three-dimensional (ISO-C3D) navigation	43	38	14 ± 1	19 ± 1	<0.001
	Pieske et al. [21]	CT-guided sacroiliac percutaneous screw placement	71	-	63 ± 39	-	-
	Takeba et al. [20]	O-arm and stealthstation navigation for screw fixation	10	-	39 (25–68)	-	-
	Yang et al. [16]	3D printed external template to guide iliosacral screw insertion	22	18	18 ± 5	40 ± 11	<0.001
Operation time overall (min) Mean ± std	Cai et al. [4]	3D printing-based minimally invasive cannulated screw treatment	65	72	59 ± 13	72 ± 13	<0.001
	Chen et al. [23]	Minimally invasive screw fixation using the “Blunt End” kirschner wire technique assisted by 3D printed external template	28	-	85 (60–150)	-	-
	Hung et al. [3]	Pre-operative virtual simulation and 3D printing-assisted contoured plate	16	14	206 ± 70	276 ± 90	0.023
	Li et al. [19]	Computer-aided angiography and rapid prototyping technology	81	76	105 ± 19	122 ± 23	0.035
	Privalov et al. [28]	Intra-operative CT in navigated sacroiliac instrumentation	25	28	189 ± 89	C^1^: 153 ± 68 C^2^: 201 ± 100 C^3^: 127 ± 70	0.31 0.70 0.14
Blood loss (mL)Mean ± std or Mean (range)	Hung et al. [3]	pre-operative virtual simulation and 3D printing-assisted contoured plate	16	14	275 ± 197	549 ± 404	0.023
	Nie et al. [27]	3D printing assisted by minimally invasive surgery	30	-	31 ± 11	-	-
	Takeba et al. [20]	O-arm and stealthstation navigation for screw fixation	10	-	12 (0–120)	-	-
Fluoroscopy Dose mean ± SD or mean (range) presented in the given unit	Balling [6]	3D image guided sacral screw fixation via single-sided minimally invasive transgluteal approach	52	-	788 ± 632mGy/cm	-	-
	Beck et al. [22]	Intra-operative 3D fluoroscopy of iliosacral screws and lumbopelvic implants	14	12	181 cGy/cm^2^ (90–424)	1376 cGy/cm^2^ (485–2)	NA
	Ghisla et al. [25]	Intra-operative 3D-CT guided navigation for sacro-iliac screws	21	-	1918 mGy/cm	-	-
	Pieske et al. [21]	CT-guided sacroiliac percutaneous screw placement	71	-	Male: 6 ± 3 msV, range: 2–17; Female: 9 ± 3 msV, range: 1–28	-	-
	Yang et al. [16]	3D printed external template to guide iliosacral screw insertion	22	18	743 ± 231 cGy/cm^2^	1904 ± 845 cGy/cm^2^	<0.001
Fluoroscopy time (sec) mean ± SD or mean (range)	Beck et al. [22]	Intra-operative 3D fluoroscopy of iliosacral screws and lumbopelvic implants	14	12	64 (60–71)	181 (54–340)	NA
	Berger-Groch et al. [14]	3D navigated iliosacral screw placement	100	36	99 ± 812	164 ± 166	0.02
	Gao et al. [24]	Minimally invasive fluoro-navigation screw fixation	22	-	22 (10–46)	-	-
	Kim et al. [26]	Percutaneous iliosacral screwing using 3D-fluoroscopy	29	-	84 (22–160)		
	Li. et al. [17]	Percutaneous screw fixation using three-dimensional (ISO-C3D) navigation	43	38	34 ± 2	58 ± 5	<0.001
	Privalov et al. [28]	Intra-operative CT in navigated sacroiliac instrumentation	25	28	82 ± 97	C^1^: 299 ± 374 C^2^: 243 ± 92 C^3^: 248 ± 191	0.03 0.00 0.02
Fluoroscopy frequency number of times in mean ± SD or mean (range)	Cai et al. [4]	3D printing-based minimally invasive cannulated screw treatment	65	72	29 ± 4	37 ± 3	<0.001
	Chen et al. [23]	Minimally invasive screw fixation using the “Blunt End” kirschner wire technique assisted by 3D printed external template	28	-	35 (28–60)	-	-
**Post-Operative Results**							
Screw malposition rate (%)	Beck et al. [22]	Intra-operative 3D fluoroscopy of iliosacral screws and lumbopelvic implants	14	12	7	6	NA
	Gao et al. [24]	Minimally invasive fluoro-navigation screw fixation	22	-	2	-	-
	Ghisla et al. [25]	Intra-operative 3D-CT guided navigation for sacro-iliac screws	21	-	3	-	-
	Kim et al. [26]	Percutaneous iliosacral screwing using 3D-fluoroscopy	29	-	23	-	-
	Li. et al. [17]	Percutaneous screw fixation using three-dimensional (ISO-C3D) navigation	43	38	5	24	0.015
	Pieske et al. [21]	CT-guided sacroiliac percutaneous screw placement	71	-	1	-	-
	Takao et al. [15]	3D fluoroscopic navigation of iliosacra screw insertion	27	-	7	-	-
	Takeba et al. [20]	O-arm and stealthstation navigation for screw fixation	10	-	0	-	-
	Teo et al. [18]	Sacroiliac screw placement with and without intra-operative navigation	17	19	12	5	0.48
	Yang et al. [16]	3D printed external template to guide iliosacral screw insertion	22	18	3	14	<0.001
	Berger-Groch et al. [14]	3D navigated iliosacral screw placement	100	36	14	21	0.09
Reduction according to Matta (excellent + good in %)	Cai et al. [4]	3D printing-based minimally invasive cannulated screw treatment	65	72	79	81	0.762
	Chen et al. [23]	Minimally invasive screw fixation using the “Blunt End” kirschner wire technique assisted by 3D printed external template	28	-	89	-	-
	Nie et al. [27]	3D printing assisted by minimally invasive surgery for pubic rami fractures	30	-	100	-	-
	Yang et al. [16]	3D printed external template to guide iliosacral screw insertion	22	18	86	89	1.000
Functional outcome (Majeed excellent + good rate in %)	Cai et al. [4]	3D printing-based minimally invasive cannulated screw treatment	65	72	82	81	0.884
	Chen et al. [23]	Minimally invasive screw fixation using the “Blunt End” kirschner wire technique assisted by 3D printed external template	28	-	82	-	-
	Li. et al. [17]	Percutaneous screw fixation using three-dimensional (ISO-C3D) navigation	43	38	92	89	0.637
	Nie et al. [27]	3D printing assisted by minimally invasive surgery for pubic rami fractures	30	-	100	-	-

NA, not addressed.

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
