# Peer review of "Does 3D-Assisted Operative Treatment of Pelvic Ring Injuries Improve Patient Outcome?—A Systematic Review of the Literature"

_jpm, 2021, doi:10.3390/jpm11090930_

Round 1

Reviewer 1 Report

The topic sounds very interesting, also considering that 3D methodologies are acquiring an increasingly large role in the surgical field. The methodology for the selection of the articles considered in the review seems to be robust. Notwithstanding, some points seem to need clarification:

  • Authors established a quality ranking for the considered articles, but after that they treated all articles as having the same quality. Could authors add some consideration in this respect?
  • Considering the number of patients considered in the review, 694 have been treated with a 3D approach and 294 have been treated with a conventional one. Could the authors add some comments on the possible effects (if any) of this imbalance?
  • The typology of 3D approaches and of the article types considered seems to be very heterogeneous (five applications and just eighteen articles, eight of which are considered by the authors to be poor quality): is it sufficient to draw even indicative conclusions? Could the authors add some comments on this regard?

Author Response

Response to reviewer 1

  1. Authors established a quality ranking for the considered articles, but after that they treated all articles as having the same quality. Could authors add some consideration in this respect?
    Answer: we agree that we treated all articles as having the same quality and that this is not an optimal situation in the case of our review with many different quality studies. Therefore, we added an additional best-evidence synthesis analysis in order to balance the results. All the adjusted parts in the manuscript can be found right here below, highlighted in yellow.

Abstract
Methods. [….] Differences in operation time, blood loss, fluoroscopy time, screw malposition rate, fracture reduction and functional outcome between 3D-assisted and conventional (2D) pelvic injury treatment were evaluated and a best-evidence synthesis was performed.’

Methods section
‘[…] Instead, weighted means of the various outcome variables of the studies were calculated for comparison. Besides, a best-evidence synthesis was performed, taking into account the methodological quality and outcome of the original studies (table 4) (1,2). Excellent and good quality studies were labeled as high-quality studies whereas moderate and low-quality studies were labeled as low-quality.

Table 4. Best-evidence synthesis

Strong evidence

Consistent findings among multiple high-quality studies

Moderate evidence

Consistent findings in multiple low-quality studies and/or one high-quality study

Limited evidence

Consistent findings in at least one low-quality study

Conflicting evidence

Inconsistent findings among multiple studies (high- and/or low-quality studies)

No evidence

Findings of eligible studies do not meet the criteria for one of the levels of evidence stated above, or there are no eligible studies available

Results section
‘Best-evidence synthesis
Intra-operative results
Compared to conventional pelvic ring injury surgery, moderate evidence was found for a decrease in operation time per screw, operation time overall, blood loss, fluoroscopy dose and fluoroscopy time.  The evidence for a decrease in fluoroscopy frequency was limited. Conflicting evidence was found for a decrease in screw malposition rate.

Post-operative results
Moderate evidence was found that fracture reduction as well as functional outcome did not improve using 3D assisted pelvic ring injury surgery.’

Discussion section
‘Limitations of this systematic review are considered small patient groups and a wide range of methodological quality including a substantial number of moderate and poor-quality studies. Hence, a best-evidence synthesis was performed which is a transparent and commonly applied method  attempting to answer the key questions (1,2)’.

  1. Considering the number of patients considered in the review, 694 have been treated with a 3D approach and 294 have been treated with a conventional one. Could the authors add some comments on the possible effects (if any) of this imbalance?
    Answer: We agree that there is an unequal number of patients in the conventional group compared to the 3D group. However, nine out of the 18 studies concerned case-control studies. As these studies compared results after conventional 2D techniques with 3D assisted techniques, no imbalance is expected in these studies. In order to address the imbalance in the total study population, we calculated weighted means of the conventional and 3D results separately, and thus corrected for the number of patients included in the studies. Therefore, we believe that the risk of bias due to an imbalance in the overall number of patients treated with a 3D approach or a conventional approach is limited and won’t affect conclusions.
  2. The typology of 3D approaches and of the article types considered seems to be very heterogeneous (five applications and just eighteen articles, eight of which are considered by the authors to be poor quality): is it sufficient to draw even indicative conclusions? Could the authors add some comments on this regard?
    Answer: We agree with the reviewer that the included articles are heterogeneous in different ways and that we should be cautious with drawing our conclusions based on the results. A part of this remark has been addressed to in the answer to the reviewer’s first remark, concerning the poor-quality studies, for which we additionally performed a best synthesis analysis. Concerning the intra- and postoperative outcomes, there was a large difference in the number of studies that evaluated the various outcome measures. For example, blood loss was analyzed in three studies and fluoroscopy frequency in two studies. Therefore, we also used weighted means, thereby correcting for the number of patients for each of the outcome measures. Moreover, the heterogeneity in terms of patient numbers, methodological quality and different outcome variables used was mentioned in the limitation section. As a result, conclusions based on the results of this study should be indeed interpreted with caution. Therefore, we adjusted the conclusion section of the abstract, as well as the conclusion of the article, which can be found below. Still, we believe that this study provides the first complete overview of all state-of-the-art 3D applications for pelvic ring injury surgery published over the last decade.

Abstract
Conclusion.
[…] Due to a wide range of methodological quality and heterogeneity between the included studies, results should be interpreted with caution.

Conclusion
‘[…] Due to the heterogeneity of the included studies in terms of methodological quality and number of studies that evaluated each of the outcomes of interest, results should be interpreted with caution. Future high-quality comparative studies are necessary to further establish possible advantages of 3D-assisted surgery in the treatment of pelvic ring injuries.’

We hope to have answered all of the reviewers’ questions and managed to correctly address all of the remarks in the revised manuscript. We hope this will enable publication. 

Kind regards,
Hester Banierink, MD

Reviewer 2 Report

The manuscript provides a comprehensive overview of existing research on the use of 3D assisted operative treatment of pelvic ring injuries, with the goal of assessing if the assistive methodology improved the patients experience based on several quantitative factors (blood loss, operation time, etc.) intra- and post-operatively. The manuscript is detailed and well structured, however there a couple of things to be addressed before publication:

1 Abstract’s methods.  It looks like there were very particular metrics that were tracked - such as operation time, blood loss, implant malposition, etc. However these should be listed in the methods in a concise manner. They appear in the results, without being mentioned in the methods.   Also, the term “methodological quality” has a very specific definition (explained in the methods section of the text body) within this manuscript, however, without knowledge of this specific definition, the reader might misunderstand it when only reading the abstract. In other words, I am not sure if it is common knowledge to know its local definition, while the commonly know meaning of methodological quality is very general and might mean different specific things across disciplines. Consider rewording this part of the abstract.

  1. Page 3. HB, FIJ - it’s a bit unusual to refer to the specific authors that did a particular task as part of the work for the transcript within the manuscript itself. This is already handled by a dedicated section at the end of the manuscript. Is this needed in the text?
  2. Page 5 top. RCT appears to not have been defined previously in the text.
  3. Overall, there needs to be a more detailed description of the meaning and the reason for choosing each of the quantitative factors for assessing the performance of each assistive methodology – blood loss, operation time, etc. Why were they chosen, and were there any other factors considered in literature and why not here? This should be clarified in the methodology in a bit more detail and with some references for justification for the choice of each factor – regardless of how obvious some might be.

Author Response

Response to reviewer 2

  1. Abstract’s methods.

(1.) It looks like there were very particular metrics that were tracked - such as operation time, blood loss, implant malposition, etc. However these should be listed in the methods in a concise manner. They appear in the results, without being mentioned in the methods.   

(2.) Also, the term “methodological quality” has a very specific definition (explained in the methods section of the text body) within this manuscript, however, without knowledge of this specific definition, the reader might misunderstand it when only reading the abstract. In other words, I am not sure if it is common knowledge to know its local definition, while the commonly known meaning of methodological quality is very general and might mean different specific things across disciplines. Consider rewording this part of the abstract.
Answer: We will address the remarks separately:
                1. Metrics: We agree with the reviewer and added the metrics that were studied to the methods section of the abstract. You can find the adjusted section below, highlighted in yellow.

‘Methods. A systematic review was performed including published studies between January 1st 2010 and May 22nd 2020 on all available 3D techniques in pelvic ring injury surgery. Studies were assessed for their methodological quality according to the Modified McMaster Critical Review form. Differences in operation time, blood loss, fluoroscopy time, screw malposition rate, fracture reduction and functional outcome between 3D-assisted and conventional (2D) pelvic injury treatment were evaluated.’

                2. Methodological quality: in the context of a systematic review, we believe that methodological quality is a quite common term, although it can be assessed with different instruments. Because we included studies of all types, we found that the Modified McMaster
                The critical Review form was the most appropriate one to use. As the abstract is limited by the number of words that is allowed, we found it hard to provide a detailed description of the instrument that was used. However, we have added the name of the instrument we used in the methods section.

  1. Page 3. HB, FIJ - it’s a bit unusual to refer to the specific authors that did a particular task as part of the work for the transcript within the manuscript itself. This is already handled by a dedicated section at the end of the manuscript. Is this needed in the text?
    Answer: We agree with the reviewer, we removed the initials from the manuscript

  2. Page 5 top. RCT appears to not have been defined previously in the text.
    Answer: Indeed, we didn’t mention the abbreviation in the methods section and clarified this part.

[…] A total of 18 articles were included in the review of which most were case-control studies (N=9), followed by cross-sectional cohort studies (N=8) and one prospective cohort study. No randomized controlled trials (RCT’s) were found in this search.

  1. Overall, there needs to be a more detailed description of the meaning and the reason for choosing each of the quantitative factors for assessing the performance of each assistive methodology – blood loss, operation time, etc. Why were they chosen, and were there any other factors considered in literature and why not here? This should be clarified in the methodology in a bit more detail and with some references for justification for the choice of each factor – regardless of how obvious some might be.
    Answer: We agree with the reviewer that it would be helpful to provide a more detailed description of the reason for choosing each of the factors. These quantitative factors were chosen because these are the most frequently used objective outcome measures for the evaluation of pelvic ring surgery. Operation time, blood loss, and fluoroscopy should be considered parameters that represent efficiency of the operation. We hypothesized that meticulously 3D surgical planning could affect efficiency. Therefore, these parameters were chosen as outcome measures. Moreover, screw malposition rate, fracture reduction and functional outcomes are frequently used outcome parameters in literature about pelvic ring surgery. These parameters represent the accuracy of the operative procedure itself. As suggested by the reviewer, we searched for key papers about pelvic ring injuries and our search reconfirmed that these parameters are used for assessment of pelvic ring surgery. We hypothesized that 3D surgical planning might affect the accuracy of screw placements and fracture reduction. Therefore, these parameters were chosen as outcome measures for our review. Also, improved fracture reduction might in the end affect functional outcome. Hence, we believe these are the most widely known and studied outcome measures. To the best of our knowledge, there are no other factors considered in literature. We added some references to justify our choice for each factor. Therefore, we made some adjustments to the methods section:

‘Outcomes directly related to the operative treatment should be reported. These included operation time, blood loss, screw malposition rate and fluoroscopy time or fluoroscopy frequency, fracture reduction and functional outcome. These outcome measures represent the efficiency and accuracy of the surgical procedure itself. We hypothesized that 3D assisted surgery could have an effect on these measures, which is the rationale to choose these outcome measures. Moreover, these are widely used for assessing pelvic ring surgery related to patient outcomes (Zwingmann et al, 2013; Pastor et al, 2019; Banierink et al, 2020).’

Referenced to be added:

  1. Regarding the outcome screw malposition - Zwingmann J, Hauschild O, Bode G, Südkamp NP, Schmal H. Malposition and revision rates of different imaging modalities for percutaneous iliosacral screw fixation following pelvic fractures: a systematic review and meta-analysis. Arch Orthop Trauma Surg. 2013 Sep;133(9):1257-65.
  2. Regarding fracture reduction – Pastor T, Tiziani S, Kasper CD, Pape H-C, Osterhoff G. Quality of reduction correlates with clinical outcome in pelvic ring fractures. Injury. 2019 Jun;50(6):1223-1226. doi: 10.1016/j.injury.2019.04.015. Epub 2019 Apr 22.
  3. Regarding functional outcome - Banierink H, Ten Duis K, Wendt K, Heineman E, IJpma F, Reininga I. Patient-reported physical functioning and quality of life after pelvic ring injury: A systematic review of the literature. PLoS One. 2020 Jul 17;15(7):e0233226. doi: 10.1371/journal.pone.0233226.  eCollection 2020.

We hope to have answered all of the reviewers’ questions and managed to correctly address all of the remarks in the revised manuscript. We hope this will enable publication.

Kind regards,

Hester Banierink, MD
